# Fed-GraB: Federated Long-tailed Learning with Self-Adjusting Gradient Balancer

**Zikai Xiao**[1,*], **Zihan Chen**[2*] **Songshang Liu**[1] **Hualiang Wang**[3] **Yang Feng**[4]
**Jin Hao**[5] **Joey Tianyi Zhou**[6] **Jian Wu**[1] **Howard Hao Yang**[1†]
**Zuozhu Liu**[1†]
[1]Zhejiang University,
[2]Singapore University of Technology and Design,
[3]The Hong Kong University of Science and Technology,
[4]Angelalign Technology Inc,
[5]State Key Laboratory of Oral Diseases, Sichuan University,
[6]Centre for Frontier AI Research (CFAR), A*STAR , Singapore
zikai@zju.edu.cn

## Abstract

Data privacy and long-tailed distribution are the norms rather than the exceptions in many real-world tasks. This paper investigates a federated long-tailed learning (Fed-LT) task in which each client holds a locally heterogeneous dataset; if the datasets can be globally aggregated, they jointly exhibit a long-tailed distribution. Under such a setting, existing federated optimization and/or centralized long-tailed learning methods hardly apply due to challenges in (a) characterizing the global long-tailed distribution under privacy constraints and (b) adjusting the local learning strategy to cope with the head-tail imbalance. In response, we propose a method termed `Fed-GraB`, comprised of a Self-adjusting Gradient Balancer (SGB) module that re-weights clients' gradients in a closed-loop manner based on the feedback of global long-tailed prior derived from a Direct Prior Analyzer (DPA) module. Using `Fed-GraB`, clients can effectively alleviate the distribution drift caused by data heterogeneity during the model training process and obtain a global model with better performance on the minority classes while maintaining the performance of the majority classes. Extensive experiments demonstrate that `Fed-GraB` achieves state-of-the-art performance on representative datasets such as CIFAR-10-LT, CIFAR-100-LT, ImageNet-LT, and iNaturalist. Our codes are available at https://github.com/ZackZikaiXiao/FedGraB.

## 1 Introduction

Federated learning (FL) is an approach for massively distributed clients to train a global machine learning model collaboratively without exposing their private data, which has garnered ever-increasing attention in academia and industry alike [1, 2]. Unlike conventional machine learning methods, FL brings the learning objective directly onto the end-user devices for local training, where only the intermediate parameters (e.g., gradients) need to be sent to a server for model aggregation and update. This approach not only substantially reduces the communication overheads but, more importantly, facilitates the clients in obtaining a generic global model without sharing their private data, thereby contributing to the development of trustworthy intelligent systems [3]. Despite its great potential,

---

[*]Co-first author.
[†]Co-Corresponding author.

37th Conference on Neural Information Processing Systems (NeurIPS 2023).

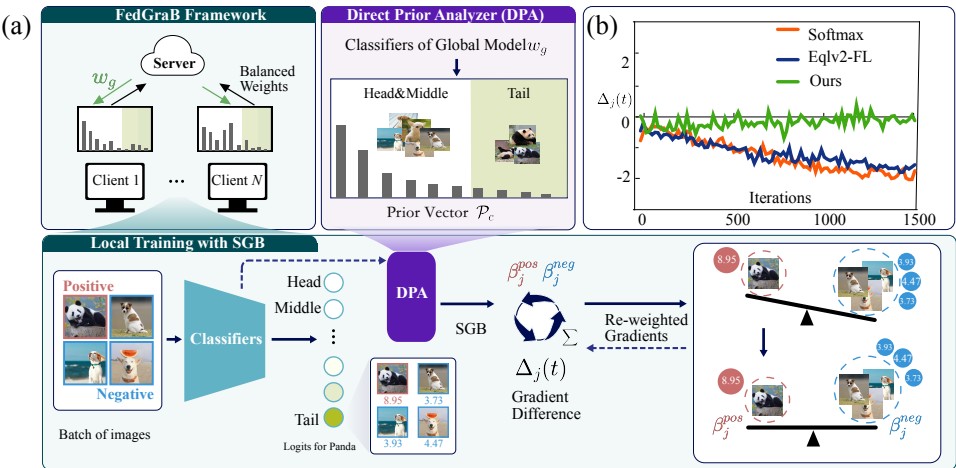

Figure 1: (a): An illustration of the `Fed-GraB` framework, in which SGB is mounted on all classes of each client based on the prior vector $\mathcal{P}_c$ derived by the DPA module, ensuring federated balanced local training; (b): Comparison for the difference of positive and negative gradients.

realizing FL also requires one to overcome new hurdles in real-world implementations, among which a particular challenge stems from data heterogeneity, i.e., the non-IID distribution and imbalanced dataset sizes across clients [4–7]. These factors often give rise to a pernicious issue where the data set of different clients jointly follows a global long-tailed distribution, which is commonly known as the Federated Long-Tailed learning (Fed-LT) problem [8, 9]. For instance, patients' diagnosis varies substantially across medical centers but collaboratively form long-tailed distributions for certain diseases [10, 11]. In a broad range of applications, performance on the minority classes (i.e., tail classes) [12], such as rare diseases, dangerous behaviors in autonomous driving [13, 14], and abnormal breath or heart rates in wearable devices [15], play a pivotal role in developing reliable and robust solutions to an FL system.

The difficulty in addressing the Fed-LT problem primarily stems from the fact that although clients' datasets can be locally balanced, globally, i.e., if the datasets were aggregated, the distribution may be long-tailed. As clients would not disclose their data information due to privacy concerns, identifying the global long-tailedness in data distribution could be onerous. To that end, a natural question arises:

Question 1: *How to leverage the global long-tailed statistics, especially for the minorities that need to be carefully tailored, without conflicting with privacy concerns?*

On the other hand, even if the globally long-tailed data information is available, existing methods do not directly apply to address such a problem. Specifically, typical FL methods cope with data heterogeneity via generic techniques such as dynamic regularizations, client selections, data augmentation, distillation, and personalization training [16–19, 4, 20, 21]. These approaches ignored the diverged/imbalanced class levels that can result in the global long-tailed issue [22, 23]. Hence, they cannot guarantee decent performance for the tail classes. A key ingredient to improve long-tailed learning is to boost the performance of tail

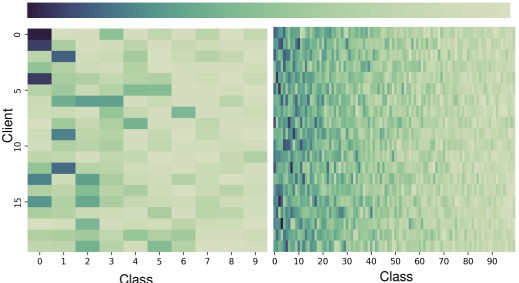

Figure 2: An example of the global and local distributions of CIFAR-10-LT (Left) and CIFAR-100-LT (Right) under non-IID setting. The bar on the top demonstrates the sorted global long-tailed distribution. Each row represents the diverse statistics of class imbalance of each client.

classes without degrading the performance of the head classes [24, 25]. While several learning methods dedicated to long-tailed data exist, they usually assume global class priors for re-balancing, which are not available in the context of FL due to privacy concerns [8]. Moreover, these methods may not be applicable in local training as local distributions could present diverse long-tailed characteristics and/or are not even long-tailed [9]. As such, it begs another question as follows:

`Question 2:` *How to perform local training to synergistically aggregate a global model that excels on both majority and minority classes under the FL setting?*

A possible choice is to re-balance the clients' local gradients via approaches such as Seasaw loss [26] or Equalization Loss v2 [27]. However one needs to customize these methods on different clients with heuristically selected hyper-parameters, resulting in complicated models with limited generalization capability. Moreover, the trained local models may have huge variance, as illustrated in Fig. 4 (a), and hardly formulate a synergistic model to handle tail class.

In light of the above challenges, this paper proposes `Fed-GraB`, comprised of a long-tailed statistic analyzer and a gradient rebalancer, for the Fed-LT problem, as illustrated in Fig. 1. More specifically, we develop a federated weight-norm-based method, coined as the Direct Prior Analyzer (DPA), to derive a prior vector for global data statistics, utilizing the weight parameters from the global classifier. To overcome the degradation caused by the distribution discrepancy between the server and clients (or amongst the clients) in Fed-LT, we establish a self-adjusting gradient balancer (SGB) based on the estimated global head-tail characteristics, which is integrated at every client to re-balance the positive and negative gradients in a class-wise and privacy-preserving manner. A marked advantage of `Fed-GraB` is that it trains all heterogeneous clients in a *feedback-based closed-loop* manner, encouraging all clients to contribute to the global long-tailed recognition task collaboratively. We also provide comprehensive theoretical and empirical analyses to verify the effectiveness of `Fed-GraB`. We conduct excessive experiments on multiple benchmark datasets (including the CIFAR-10/100-LT, ImageNet-LT, and iNaturalist) with both long-tailed and non-IID data distributions. The results demonstrate that `Fed-GraB` outperforms state-of-the-art (SOTA) baselines, including federated optimization for heterogeneous data, FL methods for LT classification, and centralized LT learning methods in FL settings.

## 2 Related Work

### 2.1 Federated learning with data heterogeneity

Numerous FL approaches have been developed to cope with the heterogeneous or imbalanced data distribution. For instance, `FedProx` [16], `FedDyn` [17], and `MOON` [28] modify the local loss function by adding regularizers at the local side. SCAFFOLD [18] proposes a control variates-based method to reduce the clients' distribution drift brought by the discrepancies. FedIR [4] dynamically adjusts the sampling ratio of each client based on each client's contribution and data distribution situation. And FedNova [20] is proposed to tackle the induced objective inconsistency problem. On the server side, `FedAvgM` [19] and `FedAdaGrad` [29] are proposed to mitigate the performance degradation. Moreover, FL-oriented client selection [30, 31] and data augmentation strategies [32] have been investigated to enhance the performance. To address the poor generalization of a single generic model, personalization training methods have been recently explored to maintain multiple personalized local models [33, 34, 21]. However, most of these methods pay more attention to the discrepancies in inter-client distributions while ignoring the inconsistency among different classes, failing to achieve satisfactory performance on the tail classes. In lieu of this, an alternate paradigm [35] solely addresses the personalized long-tail conundrum, disregarding the variances within inter-client distributions. Notable exceptions could be found in [8, 36, 6] to tackle class imbalance. Still, they require the exchange of local features [8], private local data distribution [36], and auxiliary data [6], which would lead to potential privacy issues.

### 2.2 Long-tailed learning

Long-tailed data widely exists in real-world machine learning tasks, where head classes dominate the training [10, 37]. To tackle the poor performance of tail classes in such scenarios, many approaches have been explored to boost the performance of tail classes via class re-balancing, information augmentation, and module improvements [38–41, 25]. Re-balancing strategies, as the mainstream approaches in long-tailed learning, can be categorized into class re-sampling, re-weighting, and logit adjustments [10, 42]. Re-sampling approaches [43–46] address the class imbalance via over-sampling tail classes or under-sampling head classes. On the other hand, the re-weighting approaches aim to balance the loss( [47–50]) or gradients (Seesaw loss [26], Eqlv2 [27]). Specifically, Focal loss [39] increases the prediction probabilities to achieve better prediction performance for tail classes. And a label-distribution-aware margin loss LDAM is introduced by [49] using label frequency. Moreover, a

multi-expert framework (TADE [51]), prototype learning (OLTR [24]), and head-to-tail knowledge transfer (LEAP [52]) have been proposed for improved long-tailed data performance. However, such methods' effectiveness may be constrained in Fed-LT due to discrepancies in local and global statistics.

## 3 Method

### 3.1 Preliminaries

**Federated Learning:** Consider an FL system with $N$ clients and an $M$-class visual recognition dataset $\mathcal{D} = \{\mathcal{D}_k\}_{k=1}^N$ for classification tasks. $\mathcal{D}_k$ denotes the local dataset of client $k$ with size $|\mathcal{D}_k|$. Usually, the FL training process iterates multiple communication rounds until convergence. In a typical communication round $l$, the central server randomly selects a subset of clients $\mathcal{S}_l$ and distributes the latest global model $w^l$ to them. For client $k \in \mathcal{S}_l$, it performs local update to $w^l$ based on $\mathcal{D}_k$, where the locally computed model is denoted as $w_k^l$. At the end of the round, the global model could be updated by the aggregation criterion (e.g., FedAvg) for the next round computation.

**Long-tailed distribution in FL:** To characterize the long-tailed data distribution in Fed-LT, we measure the imbalance factors in global perspectives, which is denoted by $\text{IF}_G$ and computed over the global dataset $\mathcal{D}$. We could employ a predefined imbalance $\text{IF}_G$ to sample from balanced datasets, constructing long-tailed datasets; see Fig. 2 for visualization.

**Positive and Negative Gradients:** Notably, positive and negative cumulative gradients have an important effect on long-tailed tasks such as visual recognition [53], object detection [27], and instance segmentation [26]. Suppose we have a neural network-based classification model with Cross Entropy loss as $\mathcal{L}(\boldsymbol{z}) = -\sum_{i=1}^M y_i \log(\sigma_i)$, with $\sigma_i = \frac{e^{z_i}}{\sum_{j=1}^M e^{z_j}}$, where $y_i$ is the ground truth label, $\boldsymbol{z} = [z_1, z_2, \ldots, z_M]$ are the logits, and $\boldsymbol{\sigma} = [\sigma_1, \sigma_2, \ldots, \sigma_M]$ denotes the probabilities of the classifier. Given a sample with label $j$, the positive gradients are defined as the derivative of the loss with respect to $z_j$, while the rest shall be the negative gradients, i.e.,

$$\nabla_{z_j}^{\text{pos}}(\mathcal{L}) = \sigma_j, \quad \nabla_{z_{i \neq j}}^{\text{neg}}(\mathcal{L}) = \sigma_i - 1. \tag{1}$$

### 3.2 Direct Prior Analyzer (DPA)

This section details the proposed DPA module which calculates a prior vector for balanced training. It utilizes the same information required by FedAvg, without the need for extra data or distribution details. Essentially, DPA derives a prior probability vector that discerns potential tail classes by examining the L2-norm of the weight parameters within the classifiers. The reason for this design arises from existing studies [25, 54], revealing that a model recognizes head or tail classes via classifier and tends to give higher scores for head classes when trained with imbalanced data. Thus, the weight norm serves as an effective indicator of the imbalance degree since the norm of the active weights could empirically reflect the behavior and frequency of different classes. In particular, let $w_g^j$ denote the weights of classifier $j$ in the global model classifier with weights $w_g$. The estimated global distribution would be given by $\mathcal{P}_c = \{p_j \mid h(\|w_g^j\|_2)\}$, where $h(\cdot)$ is a linear transformation for scaling the norm to share of the total. This estimation can be executed after global model aggregation with full client participation.

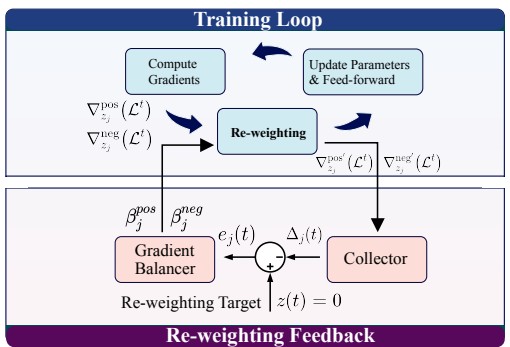

Figure 3: The closed-loop in SGB. $\beta_j^{pos}(t), \beta_j^{neg}(t)$ are updated according to the controller, and the re-weighted gradients are stored in the *collector* for future cumulative computation of $\Delta_j(t)$.

In the Fed-GraB implementation, the prior vector $\mathcal{P}_c$ derives from DPA at the onset of local training. It is also worth mentioning that $\mathcal{P}_c$ does not need to be perfect and only requires information that reflects the model prediction inclinations at the class level, rather than a complete correspondence with the distribution itself. See more detailed analysis in experiments.

## 3.3 Self-adjusting Gradient Balancer for Fed-LT

This part establishes the Self-adjusting Gradient Balancer (SGB), an effective tool that can be integrated into biased classifiers to re-balance local training and improve the overall performance. Particularly, we consider the difference between cumulative positive and negative gradients $\Delta(t)$, with $\Delta_j(t)$ for class $j$ defined as $\Delta_j(t) = g_j^{pos}(t) - g_j^{neg}(t)$, where $g_j(t) = \sum_{i=0}^{t} \nabla_{z_j}(\mathcal{L}^t)$ is the cumulative gradients of class $j$, capturing the overall imbalance degree throughout the training process. In a toy balanced scenario with $M$ samples each with equal probability to the $M$ classes, the expected target $z_j(t)$ of $\Delta_j(t)$ is:

$$E(z_j(t)) = \sum_{i=1}^{M-1} (E(\sigma_i(t)) - 1) - E(\sigma_j(t)) = 0, \tag{2}$$

indicating that $\Delta(t)$ approaches zero when the distributions become identical.

The core idea of SGB is to consistently align $\Delta_j(t)$ to 0 for tail classes in a closed-loop and self-adjusting manner. Specifically, we carry out a proportional-integral-derivative controller [55] to continually update the re-weighting coefficients of gradients with regard to the logits in a class-wise manner, as demonstrated in Fig. 3. The re-weighting process is based on adaptive self-adjustment with error feedback until $\Delta(t)$ reaches the pre-determined balanced target, rather than following heuristic methods as [26, 27]. The error feedback $e_j(t) = \Delta_j(t) - z_j(t)$ for class $j$ represents the distance between the current status $\Delta_j(t)$ and a target $z_j(t)$ during training. Given $e(t)$, the output of the controller in SGB is

$$u_j(t) = K_P e_j(t) + K_I \sum_{j=0}^{t} e_j(t) + K_D(e_j(t) - e_j(t-1)), \tag{3}$$

where $K_P$, $K_I$, and $K_D$ are controlling factors. The proportional item with $K_P$ shows the controller would give bigger re-weighting coefficients $\beta_j$ to compensate the imbalance for current larger errors. Meanwhile, the integral item with $K_I$ could look back on the past errors to reduce the steady-state offset which could not be handled by the proportional item. Furthermore, as computing $\beta_j$ only with the proportional and integral loop could hardly achieve steady and smooth re-weighting, we introduce the third differential item with $K_D$ to control the fluctuations based on future expectations. For instance, when $\Delta_j(t)$ is close to 0, the differential item would prohibit $u(t)$ from diverging re-weighting and vise versa. Overall, the three items work together to achieve quick, precise, and stable adjustments.

---

**Algorithm 1** Local training process of `Fed-GraB`

**Input:** $w_k^l$, local model at round $l$ for client $k$
**Output:** Updated local model $w_k^{l+1}$
     **function** CLIENTUPDATE($w_k^l$)
1:  Update prior vector $\mathcal{P}_c$ by DPA with $w_k^l$
2:  **for** batch $b \in \mathcal{B}$ **do**
3:     **for** each classifier $z_j \in z$ **do**
       *#Initialize $\Delta_j(t) = 0$*
4:       Compute $\nabla_{z_j}(\mathcal{L}; b) = \left[\nabla_{z_j}^{pos}(\mathcal{L}; b), \nabla_{z_j}^{neg}(\mathcal{L}; b)\right]$
5:       Sample a random value $r \sim \text{Uniform}(0, 1)$
       *# Gradient Balancer*
6:       Input of SGB $e(t) \leftarrow \Delta_j(t)$
7:       Compute output of SGB $u(t)$ by Eq. (3)
8:       Compute re-weighting coefficients $\beta_j$ by Eq. (4)
       *# Gradient Collector*
9:       Update $\Delta_j(t)$ with the re-weighted gradients by Eq. (5)
10:    $w_k^{l+1} \leftarrow w_k^l - \eta \sum_{j=0}^{M-1} \beta_j \nabla_{z_j}(\mathcal{L}; b)$
11: **return** $w_k^{l+1}$

---

We first map $u_j(t)$ through a simple activation function $\phi(\cdot)$ to calculate $\beta_j^{neg} = \phi(-u_j(t))$, $\beta_j^{pos} = \phi(u_j(t))$, where $\phi(x) = \frac{\gamma}{1+\delta e^{-\varsigma x}}$. Then, we combine the global information obtained by DPA to calculate the final coefficients for positive and negative gradients per iteration as:

$$\beta_j = \mathbb{I}_{r > \mathcal{P}_c[j]} \cdot [\beta_j^{pos}, \beta_j^{neg}] + \mathbb{I}_{r \leq \mathcal{P}_c[j]} \cdot [1, 1], \tag{4}$$

where $\mathbb{I}(\cdot)$ is the indicator function. Consequently, the re-weighted gradients are computed as:

$$\begin{cases} \nabla_{z_j}^{pos'}(\mathcal{L}^t) = \beta_j^{pos}(t)\nabla_{z_j}^{pos}(\mathcal{L}^t), \\ \nabla_{z_j}^{neg'}(\mathcal{L}^t) = \beta_j^{neg}(t)\nabla_{z_j}^{neg}(\mathcal{L}^t). \end{cases} \tag{5}$$

The working mechanism of SGB at a typical client is summarized in Fig. 3. As depicted in Fig. 1 (a), SGB is implemented across all the classes in each client, with parameters adjusted in accordance with the prior vector derived by DPA. By utilizing global information, this approach encourages all clients to contribute coherently to the global model while simultaneously rectifying prediction biases arising from long-tailed datasets. As shown in Fig. 1 (b), SGB improves the balance between cumulative positive and negative gradients. See experiments and Supplementary for further analysis.

### 3.4 Algorithm Summary for `Fed-GraB`

The training pipeline of our `Fed-GraB` framework is presented in Algorithm 1, which consists of two stages. First, DPA calculates the prior vector $\mathcal{P}_c$ after receiving aggregated weights $w_k^l$. Then, SGB is applied to all the classes. The local classifier would be dynamically adjusted with weighted positive and negative gradients via a $\Delta(t)$-based closed-loop controller in accordance with prior vector $\mathcal{P}_c$ derived from DPA. The combination with SGB and DPA is detailed in Eq. (4).

### 3.5 Privacy discussions of DPA

As the local prior computation is performed at the client side, there would be a few privacy concerns with the DPA method. It is, however, noteworthy that the potential privacy issue exists in the general FL frameworks rather than specific to our proposed DPA method. For instance, gradient inversion [56] can pose a threat to almost all gradient transmission-based FL methods without any privacy-preservation techniques. As the privacy issue of the FL framework is beyond the scope of this work, we briefly include the discussion in this subsection.

## 4 Experiments

### 4.1 Experimental Setup

**Baselines:** We consider two types of SOTA baselines: (1) FL-oriented methods to tackle data heterogeneity (`FedProx` [16], `FedNova` [20]) or federated long-tailed data-oriented (`FedIR` [4] and `CReFF` [8]), and `FedAvg` is also included for reference; (2) Long-tailed learning (LT)-oriented methods ($\tau$-norm [25], `Eqlv2` [27], `LDAM` [49], `Focal-loss` [39] and `GCL-loss` [42]) applied at local training of each client. We also provide the results of the long-tailed methods in centralized learning (CL) settings as the oracle performance upper bound, including SGB.

**Datasets:** We conduct the experiments in three benchmark datasets for long-tailed classification, i.e., CIFAR-10/100-LT [57], ImageNet-LT [58]. CIFAR-10/100-LT are sampled into long-tailed distribution by exponential distribution controlled by IF [49], and we use the same configurations as in [24] for ImageNet-LT with the number of images per class ranging from 5 to 1280. To evaluate the performance on real-world data, we also conduct experiments on iNaturalist-User-160k, with 160k examples of 1,023 species classes and partitioned on the basis of iNaturalist-2017 [59].

**Federated settings:** We use non-IID data partitions for all experiments, implemented via symmetric Dirichlet distributions with concentration parameter $\alpha$ to control the identicalness of local data distributions among all the clients. We train a ResNet-18 over $N = 40$ clients on CIFAR-10-LT. ResNet-34 and ResNet-50 are used on CIFAR-100-LT and ImageNet-LT respectively with $N = 20$ clients. For iNaturalist-160k, we use the same settings as ImageNet-LT.

**Implementation of baselines:** For the sake of fairness, we keep consistent settings external for experiments. We conduct experiments using a starting model from `FedAvg`. For `CReFF`, the number of federated features is 100, we use 0.1, 0.01 as federated feature learning rate and main net learning rate respectively on CIFAR-10/100-LT. We report the official result [8] on ImageNet-LT. As $\tau$-norm is a one-shot method, we provide a pre-trained model by `FedAvg` and adjust the classifier weights on the server as post-process. `Focal-Loss` and `EQL v2` do not require distribution prior. We directly replace the local cross-entropy loss with their proposed re-balancing methods. For `LDAM`, we use the local data distribution for its local training.

### 4.2 Comparison with State-of-the-art Methods

**Evaluation on CIFAR-10-LT:**

The performance of `Fed-GraB`/SGB on diverse settings are reported in Tab. 1. Notably, `Fed-GraB` achieves the best overall accuracies on all settings, with a significant improvement on the tail classes while ensuring good performance of the head classes. The performance gain becomes more evident under extremely imbalanced data. Moreover, the SGB implemented in the CL setting (i.e., performance upper bound testing) surpasses existing SOTA long-tail methods, further demonstrating the universal effectiveness of `Fed-GraB` under different scenarios.

| Setting | Method | IF$_G$=10 | | | | IF$_G$=50 | | | | IF$_G$=100 | | | |
|---|---|---|---|---|---|---|---|---|---|---|---|---|---|
| | | Many | Med | Few | All | Many | Med | Few | All | Many | Med | Few | All |
| CL | Softmax | 0.901 | 0.879 | 0.886 | $0.893_{\pm0.003}$ | 0.908 | 0.776 | 0.742 | $0.817_{\pm0.003}$ | 0.920 | 0.745 | 0.675 | $0.767_{\pm0.001}$ |
| | Eqlv2 | 0.902 | 0.880 | 0.874 | $0.890_{\pm0.002}$ | 0.903 | 0.774 | 0.774 | $0.819_{\pm0.006}$ | 0.912 | 0.751 | 0.679 | $0.775_{\pm0.012}$ |
| | LDAM | 0.957 | 0.799 | 0.854 | $0.838_{\pm0.026}$ | 0.964 | 0.739 | 0.685 | $0.753_{\pm0.011}$ | 0.936 | 0.729 | 0.610 | $0.698_{\pm0.030}$ |
| | SGB-CL | 0.897 | 0.901 | 0.901 | $\mathbf{0.898}_{\pm0.008}$ | 0.891 | 0.817 | 0.833 | $\mathbf{0.846}_{\pm0.004}$ | 0.901 | 0.738 | 0.814 | $\mathbf{0.818}_{\pm0.003}$ |
| α=1 | FedAvg | 0.896 | 0.858 | 0.846 | $0.877_{\pm0.001}$ | 0.888 | 0.771 | 0.693 | $0.792_{\pm0.005}$ | 0.922 | 0.716 | 0.616 | $0.737_{\pm0.001}$ |
| | FedProx | 0.898 | 0.859 | 0.854 | $0.877_{\pm0.002}$ | 0.891 | 0.773 | 0.691 | $0.794_{\pm0.002}$ | 0.921 | 0.725 | 0.582 | $0.729_{\pm0.002}$ |
| | FedNova | 0.912 | 0.853 | 0.848 | $0.882_{\pm0.002}$ | 0.903 | 0.757 | 0.702 | $0.797_{\pm0.003}$ | 0.934 | 0.734 | 0.599 | $0.739_{\pm0.005}$ |
| | FedIR | 0.966 | 0.823 | 0.862 | $0.868_{\pm0.001}$ | 0.972 | 0.775 | 0.693 | $0.784_{\pm0.002}$ | 0.969 | 0.755 | 0.576 | $0.728_{\pm0.004}$ |
| | CReFF | 0.911 | 0.850 | 0.887 | $0.884_{\pm0.002}$ | 0.896 | 0.769 | 0.664 | $0.791_{\pm0.003}$ | 0.935 | 0.723 | 0.574 | $0.726_{\pm0.002}$ |
| | $\tau$-norm | 0.887 | 0.871 | 0.908 | $0.884_{\pm0.003}$ | 0.878 | 0.790 | 0.725 | $0.805_{\pm0.002}$ | 0.922 | 0.726 | 0.668 | $0.760_{\pm0.004}$ |
| | Eqlv2-FL | 0.896 | 0.852 | 0.857 | $0.878_{\pm0.003}$ | 0.886 | 0.786 | 0.690 | $0.790_{\pm0.009}$ | 0.919 | 0.704 | 0.597 | $0.729_{\pm0.003}$ |
| | LDAM-FL | 0.901 | 0.845 | 0.825 | $0.863_{\pm0.004}$ | 0.881 | 0.739 | 0.662 | $0.768_{\pm0.005}$ | 0.891 | 0.638 | 0.495 | $0.679_{\pm0.025}$ |
| | Focal-FL | 0.887 | 0.839 | 0.834 | $0.869_{\pm0.005}$ | 0.877 | 0.744 | 0.665 | $0.775_{\pm0.002}$ | 0.916 | 0.701 | 0.558 | $0.733_{\pm0.037}$ |
| | GCL-FL | 0.923 | 0.747 | 0.781 | $0.796_{\pm0.000}$ | 0.949 | 0.748 | 0.689 | $0.761_{\pm0.005}$ | 0.963 | 0.726 | 0.608 | $0.726_{\pm0.001}$ |
| | Fed-GraB | 0.886 | 0.882 | 0.893 | $\mathbf{0.885}_{\pm0.001}$ | 0.875 | 0.784 | 0.775 | $\mathbf{0.818}_{\pm0.003}$ | 0.910 | 0.698 | 0.713 | $\mathbf{0.766}_{\pm0.003}$ |
| α=0.5 | FedAvg | 0.890 | 0.864 | 0.861 | $0.876_{\pm0.002}$ | 0.865 | 0.772 | 0.685 | $0.781_{\pm0.003}$ | 0.906 | 0.720 | 0.585 | $0.719_{\pm0.005}$ |
| | FedProx | 0.883 | 0.864 | 0.863 | $0.874_{\pm0.000}$ | 0.857 | 0.776 | 0.688 | $0.782_{\pm0.001}$ | 0.892 | 0.712 | 0.564 | $0.715_{\pm0.011}$ |
| | FedNova | 0.903 | 0.856 | 0.834 | $0.877_{\pm0.002}$ | 0.888 | 0.773 | 0.679 | $0.788_{\pm0.003}$ | 0.924 | 0.739 | 0.609 | $0.739_{\pm0.004}$ |
| | FedIR | 0.955 | 0.816 | 0.870 | $0.866_{\pm0.001}$ | 0.961 | 0.771 | 0.698 | $0.781_{\pm0.001}$ | 0.976 | 0.726 | 0.562 | $0.715_{\pm0.005}$ |
| | CReFF | 0.900 | 0.838 | 0.880 | $0.877_{\pm0.004}$ | 0.878 | 0.778 | 0.664 | $0.786_{\pm0.003}$ | 0.932 | 0.699 | 0.580 | $0.718_{\pm0.003}$ |
| | $\tau$-norm | 0.883 | 0.863 | 0.880 | $0.877_{\pm0.004}$ | 0.867 | 0.759 | 0.728 | $0.795_{\pm0.003}$ | 0.962 | 0.726 | 0.611 | $0.731_{\pm0.011}$ |
| | Eqlv2-FL | 0.895 | 0.858 | 0.855 | $0.877_{\pm0.002}$ | 0.872 | 0.771 | 0.639 | $0.775_{\pm0.005}$ | 0.898 | 0.717 | 0.586 | $0.714_{\pm0.011}$ |
| | LDAM-FL | 0.885 | 0.860 | 0.848 | $0.850_{\pm0.006}$ | 0.863 | 0.742 | 0.673 | $0.709_{\pm0.037}$ | 0.877 | 0.583 | 0.490 | $0.657_{\pm0.024}$ |
| | Focal-FL | 0.868 | 0.853 | 0.858 | $0.865_{\pm0.005}$ | 0.842 | 0.780 | 0.661 | $0.767_{\pm0.005}$ | 0.883 | 0.703 | 0.581 | $0.728_{\pm0.029}$ |
| | GCL-FL | 0.902 | 0.731 | 0.813 | $0.798_{\pm0.004}$ | 0.938 | 0.725 | 0.725 | $0.768_{\pm0.001}$ | 0.954 | 0.730 | 0.623 | $0.713_{\pm0.016}$ |
| | Fed-GraB | 0.864 | 0.891 | 0.897 | $\mathbf{0.881}_{\pm0.002}$ | 0.836 | 0.790 | 0.783 | $\mathbf{0.806}_{\pm0.001}$ | 0.910 | 0.698 | 0.713 | $\mathbf{0.761}_{\pm0.008}$ |

Table 1: Test accuracies of `Fed-GraB`/SGB and SOTA methods on CIFAR-10-LT with diverse imbalanced and heterogeneous data settings. More results (e.g., DPA with CL re-weighting methods and IID settings) are in the Supplementary.

| Method | CIFAR-100-LT | | | | ImageNet-LT | | | | iNaturalist-160k | | | |
|---|---|---|---|---|---|---|---|---|---|---|---|---|
| | Many | Med | Few | All | Many | Med | Few | All | Many | Med | Few | All |
| FedAvg | 0.643 | 0.410 | 0.182 | $0.365_{\pm0.001}$ | 0.428 | 0.258 | 0.127 | $0.287_{\pm0.006}$ | 0.596 | 0.425 | 0.242 | $0.434_{\pm0.002}$ |
| FedProx | 0.639 | 0.416 | 0.181 | $0.366_{\pm0.000}$ | 0.439 | 0.268 | 0.128 | $0.292_{\pm0.002}$ | 0.582 | 0.424 | 0.241 | $0.425_{\pm0.011}$ |
| FedNova | 0.664 | 0.429 | 0.195 | $0.378_{\pm0.009}$ | 0.415 | 0.234 | 0.114 | $0.265_{\pm0.002}$ | 0.564 | 0.403 | 0.226 | $0.404_{\pm0.006}$ |
| FedIR | 0.634 | 0.410 | 0.182 | $0.364_{\pm0.000}$ | 0.388 | 0.206 | 0.074 | $0.236_{\pm0.012}$ | 0.579 | 0.387 | 0.191 | $0.396_{\pm0.003}$ |
| CReFF | 0.684 | 0.440 | 0.146 | $0.401_{\pm0.002}$ | - | - | - | $0.263_{\pm0.000}$ | - | - | - | - |
| $\tau$-norm | 0.459 | 0.362 | 0.323 | $0.368_{\pm0.003}$ | 0.347 | 0.285 | 0.260 | $0.288_{\pm0.018}$ | 0.537 | 0.433 | 0.287 | $0.434_{\pm0.003}$ |
| Eqlv2-FL | 0.652 | 0.434 | 0.198 | $0.381_{\pm0.002}$ | 0.433 | 0.262 | 0.118 | $0.281_{\pm0.006}$ | 0.570 | 0.397 | 0.376 | $0.440_{\pm0.014}$ |
| LDAM-FL | 0.639 | 0.409 | 0.168 | $0.355_{\pm0.005}$ | 0.365 | 0.216 | 0.112 | $0.242_{\pm0.002}$ | 0.560 | 0.409 | 0.244 | $0.414_{\pm0.003}$ |
| Focal-FL | 0.645 | 0.418 | 0.179 | $0.367_{\pm0.001}$ | 0.424 | 0.266 | 0.136 | $0.283_{\pm0.005}$ | 0.596 | 0.430 | 0.247 | $0.430_{\pm0.005}$ |
| GCL-FL | 0.567 | 0.346 | 0.156 | $0.301_{\pm0.012}$ | 0.317 | 0.209 | 0.124 | $0.224_{\pm0.001}$ | 0.565 | 0.375 | 0.196 | $0.388_{\pm0.007}$ |
| Fed-GraB | 0.683 | 0.553 | 0.221 | $\mathbf{0.411}_{\pm0.002}$ | 0.407 | 0.294 | 0.215 | $\mathbf{0.311}_{\pm0.003}$ | 0.527 | 0.449 | 0.372 | $\mathbf{0.451}_{\pm0.004}$ |

Table 2: Test accuracies of various methods on CIFAR-100-LT, ImageNet-LT and iNaturalist-160k with non-IID data settings.

**Evaluation on CIFAR-100-LT, ImageNet-LT, and iNaturalist-160k:** Tab. 2 shows that `Fed-GraB` consistently outperforms all baselines on test overall accuracies for CIFAR-100-LT and ImageNet-LT, with remarkable improvements on middle and tail classes. While LT-orientated methods obtain better results than the FL method in the extremely imbalanced ImageNet-LT, `Fed-GraB` can still surpass it[3]. The reason boils down to that for classification tasks with a large number of classes, local data samples for each class would be extremely small. In this case, the SGB with global prior for class-wise re-balancing is more effective in learning a better-aggregated model.

As for iNaturalist-160k, `Fed-GraB` significantly boosts the classification accuracy on tail classes while maintaining the best overall performance compared to FL methods. In comparison to LT-oriented methods, `Fed-GraB`'s SGB provides a robust mechanism against data heterogeneity, leading to an outstanding performance on both 3-shot and overall metrics.

A salient advantage of `Fed-GraB` is that it achieves universal outperformance on three large-scale or real-world scenarios, demonstrating the robustness over other methods such as `CReFF`, $\tau - norm$ or `Eqlv2-FL` which attain good performance on a specific dataset.

**Evaluation of Communication Efficiency:** Tab. 3 presents the communication efficiency of different methods trained on CIFAR-10-LT under two non-IID settings. We compare the required communica-

---

[3]The result of CReFF on ImageNet-LT are from [8] and the result on iNaturalist is not included due to memory issues in large-scale datasets.

tion rounds for the tail classes to reach 55% test accuracy. We can see from this table that `Fed-GraB` attains the best convergence rate under both settings, confirming its effectiveness on the tail classes.

## 4.3 Effectiveness of DPA and SGB

**SGB for consistent gradient re-weighting across clients:** When various clients employ a gradient re-weighting method on a specific class, the strength of compensating divergence is because of the diverse local distributions and local dataset sizes. Therefore, although traditional re-weighting methods can mitigate the imbalance, discrepancies still exist. We demonstrate the advantage of SGB for addressing this issue. Specifically, we compute the mean and standard deviation (std) of $\Delta_j(t)$ across all clients in CIFAR-10-LT for both `Eqlv2` and `Fed-GraB`, as illustrated in Fig. 4 (a).

We make two observations here. First, the mean of $\Delta_j(t)$ in Eqlv2 keeps decreasing alongside the training process while `Fed-GraB` can always align it to a near-zero value, suggesting that `Fed-GraB` can better re-weight the positive and negative gradients. Second, `Eqlv2` exhibits a large variance among clients during training. In contrast, `Fed-GraB` maintains a very small deviation, evidently demonstrating the `Fed-GraB` can consistently motivate the heterogeneous clients to converge towards a better aggregated global model.

| Method | IF$_G$=50 | | IF$_G$=100 | |
|---|---|---|---|---|
| | $\alpha = 1$ | $\alpha = 0.5$ | $\alpha = 1$ | $\alpha = 0.5$ |
| FedAvg | 96 | 122 | 88 | 192 |
| CReFF | 285 | 286 | 309 | 348 |
| Eqlv2-FL | 301 | 210 | 191 | 205 |
| LDAM-FL | 483 | 427 | - | - |
| Focal-FL | 276 | 174 | 347 | 257 |
| Fed-GBA | **59** | **63** | **95** | **122** |

Table 3: A comparison of communication efficiency on CIFAR-10, in terms of the number of required rounds, where "-" indicate the method cannot obtain the target accuracy.

**Effectiveness on global distribution and tail class estimation:** We evaluate DPA under three IF settings with two metrics: (1) the L2-distance between the prior vector $\mathcal{P}_c$ and ground truth distribution; (2) the tail identification accuracy (percentage of tail classes that are identified correctly by DPA). We plot the two metrics on a global model with FedAvg for 200 rounds in Fig. 4 (b). We can see that DPA converges after 70 rounds of training and captures more than 90% of the tail classes under different IFs. Moreover, the tail identification accuracy becomes more accurate with a higher IF, which is attributed to the extreme imbalance among classes (i.e., easier to distinguish).

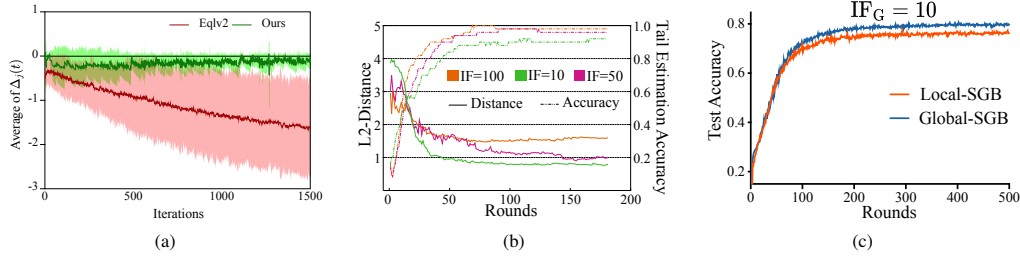

Figure 4: (a): Visualization of mean and std of $\Delta_j(t)$ during training; (b): Tail identification performance with DPA; (c): Comparison of accuracies with global/local-SGB mounting strategy.

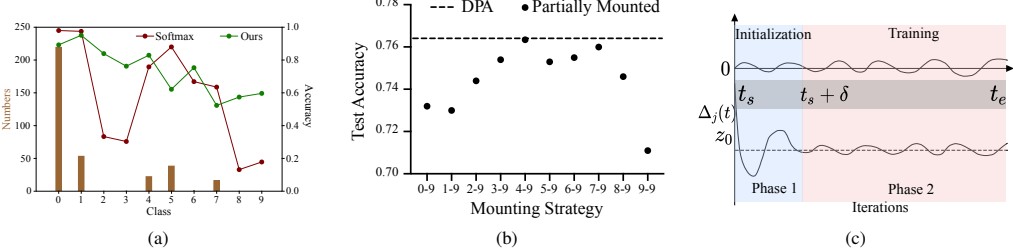

Figure 5: (a): An illustration of effectiveness of $\mathcal{P}_c$ with class-wise performance; (b): Mounting strategies with SGB; (c): An illustration of the trend of $\Delta(t)$ with different values of $z(t)$.

**Effectiveness of global $\mathcal{P}_c$:** We investigate a local version (Local-SGB) of `Fed-GraB` (Global-SGB), where each client implements SGB based on their local distribution instead of the global prior vector derived by DPA. The results on CIFAR-10-LT in Fig. 4 (c) show that Global-SGB achieves a higher

test accuracy. Furthermore, we visualize the accuracies of `FedAvg` and `Fed-GraB` after local updates on the received global model in Fig. 5 (a), which demonstrate that `Fed-GraB` can achieve better and more balanced performance across classes, especially for tails (e.g., class 2, 3, 8, 9).

**Mounting strategies with SGB:** In `Fed-GraB`, SGB is universally applied to all classifiers through the use of DPA, rather than selectively targeting specific classes without DPA. We demonstrate that a full mounting approach with DPA can more readily attain optimal performance compared to a custom-tailored tailed class (i.e., mounting on 7 to 9 classes). As depicted in Fig. 5 (b), the model excels when SGB is employed on tail classes, while accuracy diminishes when mounting all classes or solely the exceedingly biased classes. Notably, DPA significantly bolsters overall performance.

### 4.4 Ablation Study and Model Analysis

**The target of $\Delta(t)$:** A critical step of SGB is to compute re-weighting coefficients $\beta$ by $e(t) = \Delta(t) - z(t)$. Here we argue that for a static $z(t)$, different values of $z(t)$ would not lead to significant differences in training, as long as those values are not extremely biased. We visualize the training process with two different $z(t)$ in Fig. 5 (c) by assuming a continuous scenario. The training would consist of two phases during the closed-loop controlling. The *Phase 1* could be regarded as a noisy initialization stage which is usually very short, resulting in different initial values based on $e(t)$. Afterward, the *Phase 2* is started, and similar adjusting behaviors of $e(t)$ could be observed, i.e., $\Delta(t)$ fluctuates around $z(t)$, leading to similar $\beta$ that stands for re-weighting strength. As the duration $\delta$ of *Phase 1* is negligibly short and would not impose significant influence for training. The experimental results with different target values in Fig. 6 (a) further demonstrate our argument. More analysis is presented in the Supplementary.

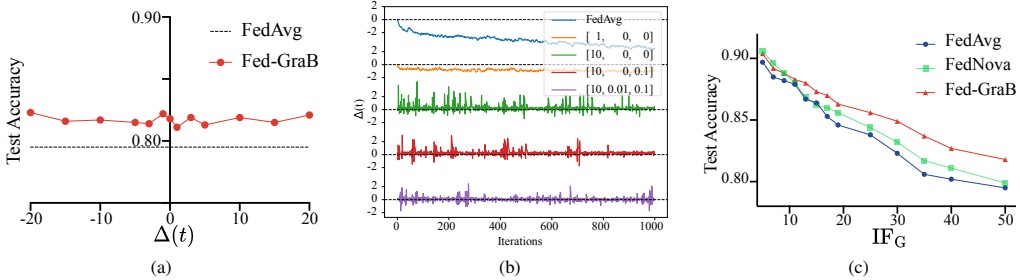

Figure 6: (a): Performance ablation with different $\Delta(t)$; (b): Ablation on different $K_P$, $K_I$ and $K_D$ in `Fed-GraB`; (c): Performance evaluation with diverse IF$_G$.

**Ablation on SGB hyper-parameters:** To investigate the effects of hyper-parameters $K_P$, $K_I$, and $K_D$, we observe the real-time $\Delta(t)$ during the training process with various parameter groups, as depicted in Fig. 6 (b) using the parameter sets: $K_P = 1, 10, 10, 10$, $K_I = 0, 0, 0, 0.01$, $K_D = 0, 0, 0.1, 0.1$. We notice that $K_P = 10$ (green) more effectively constrains $\Delta(t)$ around $0$ compared to $K_P = 1$ (orange). To reduce fluctuations, we incorporate $K_D = 0.1$ (red). In this case, $\Delta(t)$ remains above $0$, necessitating the introduction of $K_I$ (purple) to correct the static deviation. Notably, SGB demonstrates substantial robustness to the parameter group, as further evidenced in the Supplementary.

`Fed-GraB` **on different imbalance levels:** To demonstrate `Fed-GraB` is versatile across different global data distributions, we conduct experiments with global imbalance factors ranging from 5 to 50. As shown in Fig. 6 (c), `Fed-GraB` achieves better performance than others on a broad range of IF$_G$. The results indicate that `Fed-GraB` could alleviate the performance degradation in moderate imbalanced cases while yielding more significant improvements on highly imbalanced data.

**Computational and Storage Cost of SGB:**

The additional cost from SGB are mainly attributed to the computation and storage of $\Delta(t)$ and $u(t)$. As shown in Eq. (3), the main computational steps are the differential and summation of $e(t)$, which should be linearly proportional to number of the gradients. Such extra computational cost is analogous to the additional manipulations of gradients in advanced stochastic gradient decent methods such as Momentum or Adam [60]. Overall, the extra computation which is implemented with several lines of code could be done very quickly. Besides, `Fed-GraB` needs some extra storage cost to store the weighted gradients which is quite cheap as well.

# 5 Conclusions and Limitations

We proposed `Fed-GraB`, a self-adjusting and closed-loop gradient re-balancing framework for Fed-LT tasks. `Fed-GraB` comprises DPA, a federated global long-tailedness analyzer, and SGB, a local gradient balancer, addressing the discrepancies in inter-client and inter-class statistics. We carried out extensive experiments to analyze the functionality of different components in `Fed-GraB` and demonstrate the efficacy of `Fed-GraB` in various Fed-LT configurations.

**Limitations.** In this work, we mainly focus on aggregating a global Fed-LT model, while extensions to include clients are of high interest such as training superior heterogeneous clients with with personalized strategies. Besides, experiments with real-world medical or autonomous vehicle datasets could be examined to further demonstrate the effectiveness of `Fed-GraB`.

## Acknowledgements

This work is supported by the National Natural Science Foundation of China (Grant No. 62106222, No. 62201504), the Natural Science Foundation of Zhejiang Province, China(Grant No. LZ23F020008, No. LGJ22F010001), Zhejiang Lab Open Research Project (No. K2022PD0AB05) and the Zhejiang University-Angelalign Inc. R&D Center for Intelligent Healthcare.

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
