# OpenReview forum: "Fed-GraB: Federated Long-tailed Learning with Self-Adjusting Gradient Balancer"
_NeurIPS.cc/2023/Conference — NeurIPS 2023 poster_

### Official Review · Reviewer_TCDG · 2023-06-29

**Soundness:** 3 good
**Presentation:** 2 fair
**Contribution:** 3 good
**Rating:** 5
**Confidence:** 5

**Summary:**

This paper investigates a federated long-tailed learning (Fed-LT) task in which each client holds a locally heterogeneous dataset; if the datasets can be globally aggregated, they jointly exhibit a long-tailed distribution. The authors propose the Fed-GraB to coordinate the global long-tailed distribution and the local learning strategy. Specifically, it consists of a Self-adjusting Gradient Balancer (SGB) module that re-weights clients’ gradients in a closed-loop manner based on the feedback of global long-tailed prior derived from a Direct Prior Analyzer (DPA) module. Experiments are conducted on CIFAR-10-LT, CIFAR-100-LT, ImageNet-LT, and iNaturalist.

**Strengths:**

1. Investigating the federated long-tailed learning problem from the view of combining global and local sounds interesting.
2. The design of the proposed method is in line with the motivation of the paper, which seems technically feasible.
3. Experiments are conducted on many long-tailed benchmarks with various settings, which demonstrate the effectiveness of the proposed method.

**Weaknesses:**

Although the technical contribution of the manuscript sounds qualified, the presentation of the manuscript is not clear enough. There are some confusing issues that I will explain in the questions section.  Please resubmit a revised version. I will update my final score according to your revision. Meanwhile, too many hyperparameter settings will affect the generalizability of the proposed method.

**Questions:**

1. What's the correlation between uj in Eq.3  and beta_j in Eq.4?  And what's the correlation between beta_j and (beta_j^pos, beta_j^neg)? This is essential for understanding the re-weighting process of SGB.
2. In line 196, the author gives the definition of φ, but I could not find where the φ be applied.
3. In Eq.4, a random number r is adopted as a threshold for Pcj, what's the role of it in the SGB?
4. What's the definition of the expected target zj(t)? And why define the error feedback ej(t) as (gpos - gneg -  zj(t))?
5. The EQLv2 strives to keep the cumulative positive and negative gradients to be equal for each category which seems to share similar ideas with the proposed method. Compared with EQLv2-FL, what are the novelty and advantages of this approach?

**Limitations:**

The authors have declared the limitations in their paper.

---

> ### Author Rebuttal · Authors · 2023-08-10
>
> > **1. What's the correlation between uj in Eq.3 and beta_j in Eq.4? And what's the correlation between beta_j and (beta_j^pos, beta_j^neg)?Where is the φ in line 196 be applied?**
> >
>
> We apologize for any confusion caused due to the missing of certain details in the paper. We will reiterate the details here and update the revised version accordingly.
>
> The error feedback $e_j(t)=\Delta_j(t) - z_j(t)=\Delta_j(t)$ for class $j$ represents the distance between the current status $\Delta_j(t)$ and a target $z_j(t)$ during training. Hence, the input of the controller $e_j(t)$ is the cumulative difference of positive and negative gradients. Given $e(t)$, the output of the controller in SGB is $u_j(t) = K_{P}e_j(t)+K_{I} \sum_{j=0}^{t}e_j(t) +K_D(e_j(t)-e_j(t-1))$. The output $u_{j}(t)$ indicates the strength to adjust the reweighting coefficients $\beta_j$.
>
> We first map $u_j(t)$ through a simple activation function $\phi(\cdot)$ to calculate $\beta_j^{neg}=\phi(-u_j(t)),~\beta_j^{pos}=\phi(u_j(t))$, where $\phi(x)=\frac\gamma{1+\delta e^{-\zeta x}}$. Then, we combine the global information obtained by DPA to calculate the final re-weighting coefficients:
>
> $\beta_j  = [\beta_j^{\mathbf{pos}}(t), \beta_j^{\mathbf{neg}}(t)] = \mathbb{I}_{r > \mathcal{P}_c[j]} \cdot [\beta^{pos}_j, \beta^{neg}j] + $
>
> $\mathbb{I}_{r \leq \mathcal{P}_c[j]} \cdot [1, 1]$.
>
> $\mathbb{I}$ is indicator function and the random value $r$ is sampled from a uniform distribution ranging from 0 to 1, denoted as $r ∼ Uniform(0, 1)$, suggesting that tail classes $j$ with smaller priors $\mathcal{P}_c[j]$, have a higher probability of undergoing re-weighting.
>
> Afterwards, $\beta_j$ is employed to reweight the positive and negative gradients.
>
> We will update the missing part about $\phi(\cdot)$ in the revised version.
>
> > **2. In Eq.4, a random number r is adopted as a threshold for Pcj, what's the role of it in the SGB?**
> >
>
> Thanks for the comments. The value of $P_c(j)$ serves as a threshold, determining whether the SGB should operate during each local iteration. In essence, SGB is a class-wise balancer that can enhance the performance of designated classifiers. $P_c(j)$ is seen as the degree of imbalance, approximating the global distribution of class j. We've designed three deployment strategies:
>
> **S1**: SGB is applied to all classes and operates continually.
>
> **S2**: SGB is only applied to estimated tail classes and operates continually.
>
> **S3**: SGB is applied to all classes and operates based on priors derived from DPA.
>
> We discovered that **S1** can critically undermine the model due to overcorrection of the head classes, resulting in a catastrophic representation learning. As for **S2**, we observed that it facilitates model learning. Furthermore, applying SGB to different proportions of tail-end classes, as estimated by DPA, results in varying model performances (indicated by black dots, please refer to Fig. 5 (b) in the main paper). Therefore, **S2** requires the selection of an appropriate tail-end proportion. Thus, in step **S3**, there is no longer a need for the manual selection of specific-tailed classes. We utilize DPA to globally regulate the activation of SGB based on the comparison between the sampled value $r$ and $P_c(j)$, which readily achieves the optimal effect as seen in step **S2** (refer to the dashed line in Fig. 5 (b)).
>
> > **3. What's the definition of the expected target zj(t)? And why define the error feedback ej(t) as (gpos - gneg - zj(t))?**
> >
>
> The term $z_j(t)$ represents the desired target of $g_j^{pos}(t)-g_j^{neg}(t)$. During the training process, $g_j^{pos}(t)-g_j^{neg}(t)$ is adaptively constrained to stay close to $z_j(t)$. As for the desired target $z_j(t)$, in a simplified balanced scenario where there are $M$ samples each having an equal probability for the $M$ classes, the expectation of the target $z_j(t)$ for $\Delta_j(t)$ is given as: $E(z_j(t))=\sum_{i=1}^{M-1}(E(\sigma_i(t))-1)-E(\sigma_j(t))=0.$
>
> This signifies that $\Delta(t)$ approaches zero when the distributions become identical. Hence, we define $z_j(t)$ as 0, which implies the ideal balance point from the perspective of positive and negative gradients.
>
> The term $e_j(t)$ is the input to the PID controller, typically defined as the distance between the set ideal value $z_j(t)$ and the actual $g_j^{pos}(t)-g_j^{neg}(t)$. The controller adapts the positive and negative gradient weighting coefficients based on this error distance, to ensure that $g_j^{pos}(t)-g_j^{neg}(t)$ is better constrained by $z_j(t)$.
>
> > **4. Meanwhile, too many hyperparameter settings will affect the generalizability of the proposed method.**
> >
>
> We thank the reviewer for pointing out this. We agree that the presence of certain hyperparameters in our algorithm, including the internal parameters of the PID controller ($K_P, K_I, K_D$) and the parameters of the mapping function ( $\gamma,\delta,\zeta$).
>
> ***We have provided a comprehensive ablation study to discuss the impact of these six parameters in Sec. 4.2, 4.3, and 4.4 of the Supplementary Material and Sec. 4.4 of the main paper***. The conclusion drawn is that $K_P$ has a substantial impact on model performance, with a total accuracy increase of $3.4\%$ observed when $K_P$ is adjusted from $3$ to $10$. Meanwhile, other parameters demonstrate significant robustness. In addition, we carried out experiments on the CIFAR-100/10, ImageNet-LT, and iNaturalist datasets. We welcome further discussions on issues related to generalizability.
>
> > **5. The EQLv2 strives to keep the cumulative positive and negative gradients to be equal for each category which seems to share similar ideas with the proposed method. Compared with EQLv2-FL, what are the novelty and advantages of this approach?**
> >
>
> Due to space limitation, this issue has been moved to the global response. Sorry for the inconvenience.

---

> ### Comment · Reviewer_TCDG · 2023-08-15
>
> Thanks for the author's responses. Most of my concerns are addressed, therefore I improve my score slightly.

---

### Official Review · Reviewer_vVFc · 2023-06-30

**Soundness:** 3 good
**Presentation:** 2 fair
**Contribution:** 3 good
**Rating:** 5
**Confidence:** 4

**Summary:**

This paper presents an approach, Fed-GraB, for addressing the challenges of Federated Long-Tailed Learning, an issue characterized by data heterogeneity and privacy concerns. The authors tested their method on several benchmark datasets, which significantly outperforms state-of-the-art baselines. The paper is mostly an experimental work.

**Strengths:**

The proposed Fed-GraB model is based on an interesting technique called Self-adjusting Gradient Balancer to rebalance gradients. The experiments are comprehensive.

**Weaknesses:**

The authors propose to tackle the challenges posed by data heterogeneity and privacy concerns in the FL setting. However, the paper does not sufficiently justify the uniqueness of their approach. A clearer and more compelling argument is needed on how the proposed Fed-GraB model uniquely and effectively addresses these challenges.

The proposed model needs to be corroborated with theoretical analyses or at least insights, especially regarding the function and operation of the DPA modules.

The paper often assumes that global class priors are available for re-balancing, which may not always be the case in real-world applications due to privacy constraints. More importantly, it's unclear how the proposed method would handle issues where local distributions are not long-tailed or present diverse long-tailed characteristics.

A more comprehensive description of the interplay between the SGB and DPA modules, and the technical novelty there, would be helpful.

**Questions:**

Please see the above comments

**Limitations:**

more technical analysis or discussions are needed

---

> ### Author Rebuttal · Authors · 2023-08-10
>
> > **1. How the proposed Fed-GraB model *uniquely and effectively* addresses these challenges?**
> >
>
> Fed-GraB comprises two components, Self-adjusting Gradient Balancer (SGB) and Direct Prior Analyzer (DPA), each addressing distinct challenges.
>
> Challenge1: In the context of Federated long-tail learning, addressing how to estimate the global long-tailed statistics infringing on privacy concerns is a significant challenge.  ***Please note that global statistics are not available and need to be estimated through DPA***.
>
> The Direct Prior Analyzer (DPA) in the Fed-GraB model uniquely and effectively addresses the challenges in federated long-tailed learning by estimating global data statistics using the weight parameters from the global classifier. This allows the model to understand the global data distribution without transmitting additional information beyond the gradients [1].
>
> Challenge2: In addition, figuring out how to perform local training that synergistically aggregates a global model that excels on both majority and minority classes under the Federated Learning (FL) setting, poses another challenge.
>
> SGB establishes an expected target for the cumulative difference of positive and negative gradients across all clients, aiming to achieve synergistic aggregation. Instead of relying on predetermined heuristic algorithms [2-4], SGB incorporates a feedback loop for explicit reweighting and mitigates the long-tail bias of the model and enhances overall performance through the adaptive adjustment of the weighting coefficients of positive and negative gradients, mediated by a feedback loop.
>
> > **2. The paper's assumption of available global class priors for rebalancing, which may not always be applicable in real-world scenarios due to privacy constraints.**
> >
>
> Thanks for the comments. We would like to clarify that in Fed-GraB, ***we did not assume that the global prior distribution is available***, neither to the clients nor the server. Instead, we only assume that the local distributions (heterogeneous or iid) from clients would aggregate a global long-tailed distribution, while the detailed characteristics of the global LT distribution is unknown. As for the local distributions, they were obtained by Dirichlet distribution-based sampling approaches, which might be balanced or long-tailed with different numbers of data samples. The detailed partition process are in the Supplementary Section 2.3.
>
> ***As the global class prior is not available, we try to estimate the characteristics of the global distribution using the DPA module in a privacy-preserving manner.***
>
> For comprehensive contents of estimation accuracy under various imbalance factors and degrees of heterogeneity, please refer to lines 288-298 in the main paper. In real-world scenarios, we conducted experiments on real-world datasets including ImageNet-LT and iNaturalist, as demonstrated in Table 2 of the main paper.
>
> To further illustrate the accuracy of DPA estimations, we performed additional evaluations of DPA's proficiency in identifying global $50\%$ tail categories on CIFAR-100-LT. These results are recorded in the table of Question 2. The findings, as outlined in the table below, suggest that DPA offers accurate estimations across a broad spectrum of distributions.
>
> > **3. The ambiguity regarding how the proposed method would address situations where local distributions are either not long-tailed or exhibit a variety of long-tailed characteristics.**
> >
>
> Benefiting from the Dirichlet heterogeneous divisions in our work, ***the local distribution naturally encapsulates a variety of scenarios, including those that are not long-tailed and those that exhibit a range of long-tailed characteristics***. We have provided a heatmap (see Fig. 2 in the main paper), which allows for a convenient visual representation of the status of the local distribution.
>
> In the main paper, we conducted numerous experiments with various imbalance factors and alpha values (non-IID), which generate a vast array of local distribution scenarios. Please refer to Table 1 in the main paper for further details.
>
> To further demonstrate the performance of Fed-GraB across various local distributions, we also evaluated its performance under the IID condition. Specifically, we se CIFAR-10 with $IF_G=100$ and the results underscored the exceptional performance of our method under the IID condition, surpassing the current state-of-the-art by more than 1.9%.
>
> | Models | Many | Med | Few | All |
> | --- | --- | --- | --- | --- |
> | FedAvg | 0.929 | 0.720 | 0.595 | 0.733 |
> | CReFF | 0.938 | 0.734 | 0.592 | 0.738 |
> | FedNova | 0.927 | 0.736 | 0.625 | 0.749 |
> | EQLv2-FL | 0.932 | 0.734 | 0.595 | 0.738 |
> | Focal-FL | 0.929 | 0.710 | 0.573 | 0.721 |
> | Fed-GraB | 0.921 | 0.714 | 0.695 | 0.768 |
>
> > **4. A more comprehensive description of the interplay between the SGB and DPA modules is needed.**
> >
>
> SGB calculates $\beta_j^{\mathbf{pos}}(t), \beta_j^{\mathbf{neg}}(t)$. We combine global prior with SGB outputs:
>
> $\beta_j = [\beta_j^{\mathbf{pos}}(t), \beta_j^{\mathbf{neg}}(t)] = \mathbb{I}_{r > \mathcal{P}_c[j]} \cdot [\beta^{pos}_j, \beta^{neg}j] + $
>
> $\mathbb{I}_{r \leq \mathcal{P}_c[j]} \cdot [1, 1]$.
>
> $\mathbb{I}$ is indicator function and the random value $r$ is sampled from a uniform distribution ranging from 0 to 1, denoted as $r ∼ Uniform(0, 1)$, suggesting that tail classes $j$ with smaller priors $\mathcal{P}_c[j]$, have a higher probability of undergoing re-weighting.
>
>
>
> > **5. The proposed model needs to be corroborated with theoretical analyses or at least insights.**
> >
> Due to space limitation, this issue has been moved to the global response. Sorry for the inconvenience.
>
> [1] Addressing Class Imbalance in Federated Learning.
>
> [2] Seesaw Loss for Long-Tailed Instance Segmentation.
>
> [3] Fed-focal loss for imbalanced data classification in federated learning.
>
> [4] Equalization Loss v2: A New Gradient Balance Approach for Long-tailed Object Detection.

---

> > ### Comment · Reviewer_vVFc · 2023-08-16
> >
> > The reviewer has addressed most of my comments. I will increase the score.

---

### Official Review · Reviewer_nXLC · 2023-07-03

**Soundness:** 3 good
**Presentation:** 4 excellent
**Contribution:** 3 good
**Rating:** 7
**Confidence:** 4

**Summary:**

In this paper, a self-adjusting and closed-loop gradient re-balancing framework Fed-GraB was proposed and long-tailed learning tasks processing performance was improved.

**Strengths:**

This paper presents a methodology named Fed-GraB, which incorporates a Self-adjusting Gradient Balancer (SGB) module. This module dynamically adjusts the weightage of gradients from individual clients in a closed-loop manner, guided by the feedback obtained from a Direct Prior Analyzer (DPA) module. By employing Fed-GraB, clients can effectively mitigate the challenges posed by disparate data distributions during the model training process. They achieve a global model that exhibits enhanced performance on underrepresented classes while maintaining the performance levels of the majority classes.

**Weaknesses:**

1. Equation 4 lacks ending punctuation.
2. Figure 4(b) adds markers to make the curve more legible.
3. The definition of IFG is not clear. Please give a formula that says how to define IFG.
4. How to divide the dataset to get the Cifar-10/100-LT, there is no specific explanation.
5. The insight of the algorithm should be explained more clearly.
6. The baselines selected in this paper are not fully appropriate, because only one of them is designed for federated learning. The solutions to unbalanced data in FL are necessary to be compared.

**Questions:**

Please see Weaknesses

---

> ### Author Rebuttal · Authors · 2023-08-10
>
> We thank the reviewer for the careful feedback. For weakness 1 and 2, we will ensure the equation contains proper ending punctuation and markers in the revised manuscript.
>
> > **1. The definition of IFG is not clear. Please give a formula that says how to define IFG. And how to divide the dataset to get the Cifar-10/100-LT?**
> >
>
> Thank you for the comment. We apologize for the lack of clarity in the definition of IFG in the main body of the paper. The imbalance factor $IF_G$ represents the ratio of the number of samples in the head (most populous) category to the number in the tail (least populous) category within the dataset. $n_k^{(i)}$ is the number of data samples of class $i$ in client $k$. The $IF_G$ is define as $IF_G = \frac{\max_i{n^{(i)}}}{\min_i{n^{(i)}}}$, where $\begin{aligned}n^{(i)}=\sum_{k=1}^{N}n_k^{(i)}\end{aligned}$ and $N$ is the number of clients. And also, there are more details regarding $IF_G$ in our supplementary in section 2.1.
>
> In regards to dataset partition, we have provided details on how the dataset was divided to obtain Cifar-10/100-LT in the supplementary material, specifically in section 2.3. First, we truncated the dataset followed by an exponential distribution class-wisely, and controlled the imbalanced level by the global imbalanced factor. Second, we conducted sampling to divide the dataset using Dirichlet distribution-based approach for the non-IID data partition. Please refer to the supplementary material for more information.
>
> > **2. The insight of the algorithm should be explained more clearly.**
> >
>
> We appreciate your feedback and agree that the insight of the algorithm can be further explained. Fed-GraB consists of two main components: the Direct Prior Analyzer (DPA) and the Self-adjusting Gradient Balancer (SGB).
>
> 1. **Direct Prior Analyzer (DPA)**
>
>     The purpose of DPA is to ***analyze global long-tailed statistics***. It infers a prior vector of global data statistics by leveraging the weight parameters of the global classifiers. This prior vector allows us to understand the long-tailed distribution with imbalances between the head and tail.
>
> 2. **Self-adjusting Gradient Balancer (SGB)**
>
>     The SGB primarily functions to rebalance gradients on a per-class basis at each client, informed by the global head-tail properties estimated by the DPA. Unlike heuristic methods that can lead to client divergence, SGB harmonizes all clients towards a common rebalancing direction. This approach not only reduces statistical variance across clients (as illustrated in Fig. 4 (a) of the main paper) but also mitigates divergence. Therefore, SGB effectively ***achieves data rebalancing and divergence alleviation simultaneously***.
>
>
> We hope that this explanation provides a clearer understanding of the Fed-GraB algorithm. We stand prepared to answer any additional queries you might have on this subject matter.
>
> > **3. The baselines selected in this paper are not fully appropriate, because only one of them is designed for federated learning. The solutions to unbalanced data in FL are necessary to be compared.**
> >
>
> We categorize the baselines into two major classes. The first class encompasses imbalance federated learning methods, including CReFF[1], Fed-Focal Loss[2], and FedIR[3], as well as traditional heterogeneity methods such as FedProx and FedNova. The second class assesses the federated effects of existing long-tail methods. We have also incorporated a newly proposed method, ETF[4], which addresses federated global imbalance bias. We conducted experiments on the CIFAR-10 dataset, with an imbalance factor of 100 and a heterogeneity parameter alpha set to 0.5. The results are compiled and presented in the corresponding table.
>
> | method | Many | Med | Few | All |
> | --- | --- | --- | --- | --- |
> | CReff | 0.962 | 0.726 | 0.611 | 0.731 |
> | Fed-Focal Loss | 0.883 | 0.703 | 0.581 | 0.728 |
> | FedIR | 0.976 | 0.726 | 0.562 | 0.715 |
> | ETF | 0.499 | 0.624 | 0.703 | 0.631 |
> | Fed-GraB | 0.910 | 0.698 | 0.713 | 0.761 |
>
> [1] Federated Learning on Heterogeneous and Long-Tailed Data via Classiﬁer Re-Training with Federated Features.
>
> [2] Fed-focal loss for imbalanced data classification in federated learning.
>
> [3] Federated Visual Classiﬁcation with Real-World Data Distribution.
>
> [4] No Fear of Classifier Biases: Neural Collapse Inspired Federated Learning with Synthetic and Fixed Classifier.

---

> > ### Comment · Reviewer_nXLC · 2023-08-14
> >
> > Thanks for the responses. I have read the author responses as well as comments from other reviewers. The authors have provided more results and discussions regarding my concerns (the method details, insight of algorithms, and more baselines etc.). Overall, I think this paper identified an interesting Fed-LT setting, and the proposed DPA and SGB modules demonstrate to be effective with comprehensive experiments. Further extensions regarding other issues, such as privacy or theoretical analysis as other reviewer mentioned, could better enhance the quality of this paper, while the authors have provided some preliminary results and discussions in the rebuttal. Based on the overall quality of the paper/response, I’d like to keep my score.

---

### Official Review · Reviewer_XuVs · 2023-07-08

**Soundness:** 3 good
**Presentation:** 3 good
**Contribution:** 2 fair
**Rating:** 5
**Confidence:** 3

**Summary:**

This paper considers the globally long-tailed distribution of data and its impact on federated learning (FL). To address this issue, the authors propose the Fed-GraB algorithm that re-balances gradients. The problem is timely and the proposed algorithm is novel. Overall, the paper is well-written, and simulations are extensive. However, the limitations of the proposed framework are not rigorously addressed via for instance mathematical or quantitative analysis. Furthermore, the applicability of the proposed framework to various applications and other state-of-the-art FL algorithms are questionable.

**Strengths:**

The potential impact of the proposed framework would be significant, particularly on two emerging areas of research: fairness guarantee and rare-event detection. The proposed idea of re-balancing gradients is not entirely original, yet still sufficiently novel as it has yet been considered in FL for this purpose. Notably, the proposed idea does not incur significant costs in terms of computation and memory, which is a big plus for practical implementation. The literature of the considered problem and relevant frameworks is well reviewed, making the paper easily readable.

**Weaknesses:**

Some potential weakness of the proposed framework has been proactively described, yet the claims are not very convincing. In particular, in the Privacy Discussions, it claims that "potential privacy issue exists in the general FL frameworks rather than specific to our proposed DPA method...As the privacy issue of FL framework is beyond the scope of this we briefly include the discussion in this subsection." This justification hardly responds to the issues when applying the proposed framework to other privacy-preserving FL frameworks, for instance, that applies quantization and noise injection under which the proposed framework's performance may be degraded. Furthermore, in Computational and Storage Cost of SGB, it claims that "the extra computation which is implemented with several lines of code could be done very quickly. Besides, Fed-GraB needs some extra storage cost to store the weighted gradients which is quite cheap as well." This is not very convincing as it does not compare, for example, the resultant convergence speed and guarantee as well as FLOPS and bytes with the cases without the proposed solution.

**Questions:**

1. According to (1), the proposed algorithm relies on classification and cross entropy loss, as opposed to the original FL that has no restriction on its task and loss function. Is the proposed algorithm applicable to 1) non-classification tasks as well as 2) non-cross entropy loss or cross-entropy with other regularizers?

2. There are various advanced FL algorithms that intentionally distort gradients via quantization, sampling, and noise injection for compression, privacy protection, and so forth. Is the proposed algorithm still applicable to these algorithms?

**Limitations:**

Yes

---

> ### Author Rebuttal · Authors · 2023-08-10
>
> We thank the reviewer for providing thorough and insightful comments on our paper.
>
> > **1. Is the proposed algorithm applicable to non-cross entropy loss or cross-entropy with other regularizers as well as non-classification tasks?**
> >
>
> We thank the reviewer for this comment. Our proposed algorithm can be used in conjunction with different loss functions and regularization methods, as it acts on the gradients of the logits during backpropagation. During the rebuttal period, we have conducted experiment on the combination of cross entropy with L2 regularization and the focal loss. The experiments are based on the CIFAR-10 dataset, with results in the table below.
>
> As we can see, the Fed-GraB algorithm could work seamlessly with the Focal loss or the cross entropy loss with L2-regularizer, where the overall performance is even slightly improved. In contrast, the performance of original FedAvg degraded a bit when combined with the L2 regularizer. The results demonstrate that effectiveness of Fed-GraB for different loss and regularizers.
>
> | Method | Many | Med | Few | All |
> | --- | --- | --- | --- | --- |
> | FedAvg | 0.906 | 0.720 | 0.585 | 0.719 |
> | FedAvg + L2 regularizer | 0.981 | 0.778 | 0.503 | 0.709 |
> | Focal loss | 0.935 | 0.729 | 0.620 | 0.727 |
> | Fed-GraB+Focal loss | 0.958 | 0.718 | 0.640 | 0.735 |
> | Fed-GraB | 0.942 | 0.692 | 0.720 | 0.753 |
> | Fed-GraB+L2 regularizer | 0.962 | 0.667 | 0.743 | 0.757 |
>
> Regarding adaptability to non-classification tasks, theoretically, Fed-GraB can be applied to instance segmentation and detection tasks, such as Cascade Mask R-CNN on the LVIS v1 dataset. This is because Fed-GraB, similar to seesaw loss [1], EQL [2], Droploss [3] (which can be used for instance segmentation), and EQLv2 [4] (further extended to object detection), all employ the method of negative gradient over-suppression. They function by applying weighted positive and negative gradients to the classification heads in instance segmentation and object detection tasks.
>
> However, in the context of federated long-tail learning, for instance segmentation and object detection tasks, given the presence of long-tail class distribution within a single image, our dataset partitioning would require more complex and delicate design. Correspondingly, the formulation and metrics would also need to be appropriately designed. This goes beyond the scope of our current work. We believe that federated long-tail non-classification learning presents a challenging yet fruitful area with numerous application scenarios, making it a worthwhile focus for future research.
>
> > **2. How about the resultant convergence speed and guarantee as well as FLOPS and bytes with the cases without the proposed solution.**
> >
>
> Thanks for the comments. We conducted tests on CIFAR-10 with a heterogeneity of 0.5 and an imbalance factor of 100, comparing the in-process memory overhead of different baselines and our method. The number of clients was set to 40, with local epochs fixed at 5.
>
> We reported the convergence speed (how many rounds it would take to achieve a classification accuracy of 70%), the computational cost per round (how much time it would take to train the model for 1 epoch) and memory cost (expressed as a multiple of the memory consumed by the FedAvg, which is 3467MB) in the table below. Our method shows a computational cost the computation and memory nearly identical to the EQLv2, along with a much faster convergence rate. In terms of memory cost, the PyTorch gradient capture hook function does occupy some storage space. This overhead can be improved through official code refactoring.
>
> | Method | convergence speed (70%) | computational cost/round | memory cost |
> | --- | --- | --- | --- |
> | Fedavg | 280rounds | 2m 1s | 1.000x |
> | EqlV2 | 218 | 2m 34.2s | 2.023x |
> | Fed-GraB | 133 | 2m 35s | 2.024x |
>
> > **3. There are various advanced FL algorithms that intentionally distort gradients via quantization, sampling, and noise injection for compression, privacy protection, and so forth. Is the proposed algorithm still applicable to these algorithms?**
> >
>
> We have conducted experiments on CIFAR-10 with a heterogeneity of 0.5 and an imbalance factor of 100 about Fed-GraB with  the bucket_quantile algorithm of quantization[1] and the federated differential privacy algorithm (noise injection)[2]. The  performance is presented in the table below. It can be observed that distorting the gradients or inject noise tends to decrease model performance in the Fed-LT scenario. However, when compared to the FedAvg baseline, our Fed-GraB can  obtain significant performance improvements in both cases, especially in the tail classes.
>
> The results are reasonable as the current Fed-GraB frameworks are not yet tailored for gradient distortion or noise injection. We thank the reviewer for this valuable comment, and we’d like to expect future versions of Fed-GraB to better solve these issues.
>
> | Method | Many | Med | Few | All |
> | --- | --- | --- | --- | --- |
> | FedAvg | 0.906 | 0.720 | 0.585 | 0.719 |
> | Fed-GraB | 0.910 | 0.698 | 0.713 | 0.761 |
> | FedAvg + quantization | 0.969 | 0.707 | 0.472 | 0.665 |
> | Fed-GraB + quantization | 0.949 | 0.560 | 0.686 | 0.689 |
> | FedAvg + injected noise | 0.977 | 0.751 | 0.538 | 0.711 |
> | Fed-GraB + injected noise (DP) | 0.956 | 0.705 | 0.696 | 0.752 |
>
> [1] Seesaw Loss for Long-Tailed Instance Segmentation.
>
> [2] Equalization loss for long-tailed object recognition.
>
> [3] Droploss for long-tail instance segmentation.
>
> [4] Equalization Loss v2: A New Gradient Balance Approach for Long-tailed Object Detection.
>
> [5]Sketchml: Accelerating distributed machine learning with data sketches.
>
> [6] Communication-Efficient Learning of Deep Networks from Decentralized Data.
>
> [7] LDP-Fed: Federated learning with local differential privacy.

---

> > ### Comment · Reviewer_XuVs · 2023-08-14
> >
> > I have read the authors' responses to my concerns and comments. Most of them have been addressed at a satisfactory level via additional simulations. Although the baseline FedAvg is not state-of-the-art, it is still imporessive to see significantly faster convergence and lower computational complexity under various loss functions, demonstrating the huge potential of the proposed framework. Based on the overall paper quality and the aurhots' responses, I'd like to slightly increase my score.

---

### Author Rebuttal · Authors · 2023-08-10

# For Reviewer 3

> **5.The proposed model needs to be corroborated with theoretical analyses or at least insights, especially regarding the function and operation of the DPA modules.**
>

Regarding SGB, we provided the expected target $z_j(t)$ under ideal equilibrium conditions in Equation 2 of the paper. As for the error feedback regulator, our insights hail from the theory of automatic PID control within control theory, which is underpinned by a robust theoretical foundation. This includes various analyses such as Bode plots, Nyquist plots, and evaluations of phase margin and gain margin [1-5].

In the context of DPA, we explain the relationship between the Euclidean Norm of the classifier and the real distribution from two perspectives: the neural collapse and forward propagation.

From the perspective of neural collapse, four deeply interconnected phenomena occur during the training and gradual convergence of the neural network [6]. Among these, the third phenomenon (Neural Collapse 3) reveals that, up to rescaling, the last-layer classifiers, which implicitly represent the classifier decision, collapse to the class means. For balanced datasets, class means are uniformly distributed and form an equiangular tight frame simplex. However, for imbalanced datasets, class means naturally exhibit bias, and the classifier decision converges to these biased class means [7,8]. Therefore, when the model tends to predict the head classes, the class means of these classes become larger, leading to greater decision boundaries and a larger Euclidean Norm of the class vector. Numerous theories and experiments detailing this phenomenon are presented in [6,7].

From the forward propagation standpoint, the class with the maximum logit is selected as the predicted class. The logit $s_{ij}=f_iw_j^\top$ can be written as $\Vert f_i \Vert_2 \Vert w_j \Vert_2\cos\theta_{ij} \propto \Vert w_j \Vert_2\cos\theta_{ij}$, suggesting a large Euclidean Norm of $w_j$ biases the model towards predicting the corresponding class.

To validate the effectiveness of DPA, we conducted various experiments on CIFAR100-LT with different imbalance factors and heterogeneity, to demonstrate the estimation accuracy of DPA under various circumstances. The accuracy of global $50\%$ tail categories was determined as the ratio of correct predictions to the total number of predictions, with imbalance factors of 2, 50, and 100 and alpha values of 15, 0.5, and 0.05. The results are recorded in the following table. As can be seen, DPA demonstrates excellent generalizability and achieves higher accuracy under severe long-tail conditions. Additionally, we conducted experiments to illustrate the relationship between the Euclidean Norm of the classifier and the real distribution, as shown in ***Fig. 1 of the rebuttal material.***

|  | Alpha=15 | Alpha=0.5 | Alpha=0.05 |
| --- | --- | --- | --- |
| IF=2 | 0.780 | 0.740 | 0.680 |
| IF=50 | 0.960 | 0.980 | 0.860 |
| IF=100 | 0.980 | 0.980 | 0.880 |

[1] PID control system analysis, design, and technology.

[2] Tuning of PID controllers based on Bode's ideal transfer function.

[3] The direct Nyquist array design of PID controllers.

[4] Tuning of PID controllers based on gain and phase margin specifications.

[5] Performance and gain and phase margins of well-known PID tuning formulas.

[6] Prevalence of neural collapse during the terminal phase of deep learning training.

[7] Exploring deep neural networks via layer-peeled model: Minority collapse in imbalanced training.

[8] Decoupling representation and classifier for long-tailed recognition.

# For Reviewer 4:
> **5.The EQLv2 strives to keep the cumulative positive and negative gradients to be equal for each category which seems to share similar ideas with the proposed method. Compared with EQLv2-FL, what are the novelty and advantages of this approach?**
>

We thank the reviewer for pointing out this. As one of our baselines, EQLv2, developed under a centralized learning, is an effective method to alleviate the long-tail problem and has provided significant insights for the design of our methods. We will expound on the limitations of EQLv2 in the context of federated learning (FL) via two implementation methods: global EQLv2 and local EQLv2.

1.Applying local EQLv2 in FL, we noted a considerable variance in $g_j^{pos}(t)-g_j^{neg}(t)$ across different clients during the model aggregation, indicating a substantial divergence among them, which is not favourable for federated learning (please refer to Fig. 4: (a) in the main paper). SGB, on the other hand, constrains all clients' $g_j^{pos}(t)-g_j^{neg}(t)$ with the feedback loop to satisfy $E(z_j(t))=0$, thereby ensuring coordination among different clients, without the individually designing re-weighting parameters  ($\alpha, \gamma, \mu$ in EQLv2) that are suitable for each client's local distribution.

2.In the case of global EQLv2, it is necessary to upload each client's accumulated positive and negative gradients. As these accumulations closely mirror the local distributions (***refer to Fig. 2 of the rebuttal material***), this can potentially lead to privacy leakage concerns and associated risks. Conversely, SGB does not require the transmission of any extra information, thus mitigating the risks associated with privacy leakage.

In summary, while EQLv2 is an effective solution in a centralized learning environment, FedGraB has been designed for federated long-tailed learning, taking into account factors of distributed training such as self-adjusting gradients synchronization among clients and privacy preservation.

---

### Decision · Program_Chairs · 2023-09-21

**Decision:**

Accept (poster)

**Comment:**

This paper identifies a federated long-tailed learning scenario and proposes a novel method called Fed-GraB to address the corresponding challenges. It designs a Self-adjusting Gradient Balancer (SGB) module and a Direct Prior Analyzer (DPA) module that work collaboratively to resolve the Fed-LT task in a close-loop and privacy-preserving manner. The paper conducted comprehensive experiments on multiple datasets, exhibiting superior performance over extensive baselines. After the rebuttal and discussion, all reviewers  recommended accepting this paper.